# Tissue context determines the penetrance of regulatory DNA variation

Jessica M. Halow[1,2], Rachel Byron[3], Megan S. Hogan [4], Raquel Ordoñez[4], Mark Groudine [3,5], M. A. Bender[3,6], John A. Stamatoyannopoulos [1,2,7✉] & Matthew T. Maurano [1,4,8✉]

Functional assessment of disease-associated sequence variation at non-coding regulatory elements is complicated by their high degree of context sensitivity to both the local chromatin and nuclear environments. Allelic profiling of DNA accessibility across individuals has shown that only a select minority of sequence variation affects transcription factor (TF) occupancy, yet low sequence diversity in human populations means that no experimental assessment is available for the majority of disease-associated variants. Here we describe high-resolution in vivo maps of allelic DNA accessibility in liver, kidney, lung and B cells from 5 increasingly diverged strains of F1 hybrid mice. The high density of heterozygous sites in these hybrids enables precise quantification of effect size and cell-type specificity for hundreds of thousands of variants throughout the mouse genome. We show that chromatin-altering variants delineate characteristic sensitivity profiles for hundreds of TF motifs. We develop a compendium of TF-specific sensitivity profiles accounting for genomic context effects. Finally, we link maps of allelic accessibility to allelic transcript levels in the same samples. This work provides a foundation for quantitative prediction of cell-type specific effects of non-coding variation on TF activity, which will facilitate both fine-mapping and systems-level analyses of common disease-associated variation in human genomes.

[1] Department of Genome Sciences, University of Washington, Seattle, WA, USA. [2] Altius Institute for Biomedical Sciences, Seattle, WA, USA. [3] Fred Hutchinson Cancer Research Center, Seattle, WA, USA. [4] Institute for Systems Genetics, New York University Langone Medical Center, New York, NY, USA. [5] Department of Radiation Oncology, University of Washington, Seattle, WA, USA. [6] Department of Pediatrics, University of Washington, Seattle, WA, USA. [7] Division of Oncology, Department of Medicine, University of Washington, Seattle, WA, USA. [8] Department of Pathology, New York University Langone Medical Center, New York, NY, USA. ✉email: jstam@altius.org; maurano@nyu.edu

Systematic census of *cis*-regulatory elements using genome-wide profiling of DNA accessibility to the endonuclease deoxyribonuclease I (DNase I) has a critically informed understanding of tissue-specific gene regulation[1] and the genetics of common human diseases and traits[2]. But these maps provide only indirect evidence for the function of regulatory DNA and cannot address the effects of sequence variation therein. Regulatory element function depends on both genomic and cellular context, which cannot be easily recapitulated in reporter assays[3]. Profiling of DNA accessibility or protein occupancy at polymorphic sites represents a genome-scale approach to assessing local effects of regulatory variation in context[4–8]. However, this approach is limited by low sequence diversity in an individual human genome and the difficulty of accessing many disease-relevant cell types. Recognition of functional human sequence variants has thus been impeded by the lack of large-scale datasets assaying function at their endogenous context in vivo.

The laboratory mouse Mus musculus and related species have long been a key model for human disease and genome function[9,10]. Given the near-complete conservation of transcriptional regulatory machinery with humans, mouse transgenic experiments have been foundational in the understanding of human genetics and gene regulation[11,12]. The availability of mice from divergent strains/species offers a rich trove of genetic diversity dramatically exceeding that in human populations[9], and with potential access to a variety of tissues and cell types including developmental timepoints[13]. Genomic approaches have linked many of these DNA sequence changes to altered transcription factor (TF) binding[14,15], chromatin features[16,17], gene expression[18–20], and protein levels[21], and further dissection of molecular traits is highly complementary to high-throughput knockout phenotyping studies[22,23].

DNase I-hypersensitive site (DHS) maps in mouse tissues show substantial divergence in regulatory DNA compared to human DHSs[2,24], suggesting that studies of human *cis*-regulatory variation cannot directly incorporate analyses of orthologous mouse loci. Past work has shown that genetic effects on chromatin features can be modeled using TF-centric analysis[4,5]. The high conservation of *trans*-regulatory circuitry suggests that such a TF-centric approach might be able to leverage the power of mouse genetics for the interpretation of human *cis*-regulatory variation.

Here we describe high-resolution maps of allelic DNA accessibility in 4 cell and tissue types across a series of F1 hybrid mice derived from the inbred lab-derived and wild-derived strains and species. These maps reveal genetic effects on DNA accessibility which are moderated by cell and tissue context. We use these maps to derive sensitivity profiles for hundreds of TFs, facilitating the prediction of functional noncoding polymorphism across mammalian genomes. Finally, we use matching RNA-seq data to assess the correlation between accessibility and expression levels.

## Results

**Allelic analysis of DNA accessibility**. We analyzed hybrid, fully heterozygous F1 mice resulting from a cross of the reference C57BL/6J with five diverged strains or species: 129S1/SvImJ, C3H/HeJ, CAST/EiJ, PWK/PhJ, and SPRET/EiJ. We mapped DHSs in four diverse cell and tissue types, including whole kidney, liver, lung, and B cells purified from femoral bone marrow (Fig. 1a, b). We selected the highest-quality samples for deep paired-end Illumina sequencing-based on fragment length distribution (Fig. 1c, Supplementary Fig. 1) and high signal-to-noise demonstrated by a mean Signal Portion of Tags (SPOT) score of 60% (Supplementary Table 1). A total of 67 samples were sequenced to an average of 203 M reads each, including at least 2 replicates per condition (median = 3 replicates) (Supplementary

Table 1). We developed a stringent mapping procedure requiring high mappability to both the reference and a customized strain-specific genome incorporating known single nucleotide variants (SNVs) and indels[22] (Methods). Replicate samples exhibited a median correlation in DNaseI cleavage density at DHSs of 0.93 (Supplementary Fig. 2).

We identified an average of 196,276 DHS hotspots (FDR 5%) in each condition using the program hotspot2[1], and generated master lists of DHSs for each strain/cell type combination (Supplementary Table 2). Hierarchical clustering showed that samples clustered by cell or tissue type, rather than by strain (Fig. 1d), suggesting that additional strains provide access to genetic diversity while demonstrating consistent cell-type-specific regulatory landscapes.

To identify sites of allelic imbalance indicative of genetic differences affecting DNA accessibility, we developed a custom pipeline to filter and count reads mapping to each allele at known point variants in DHSs (see "Methods" section). The majority of SNVs were testable in only a single strain or cell/tissue type, suggesting that additional profiling is likely to yield further insights (Fig. 1e, f). We used a beta-binomial test to determine a statistically significant imbalance. We applied multiple testing corrections and set a significance threshold of 10% false discovery rate (FDR) and additionally required a strong magnitude of imbalance (>70% of reads mapping to one allele). Plotting the distribution of allelic ratios confirmed that our mapping strategy was not biased towards the reference allele (Supplementary Fig. 3). By pooling reads from multiple samples, we assessed imbalance on aggregate, per-cell type, per-strain, and per-sample bases (Fig. 1g). We identified a total of 13,835 strongly imbalanced SNVs out of 357,303 SNVs tested when aggregating across all samples (Supplementary Data 1). The high density of variation meant that nearly all DHSs in a given cell or tissue type harbored at least one SNV, and we were able to test for imbalance at an SNV in a median of 27% DHSs per cell or tissue type (Fig. 1h). The more highly diverged strains contributed substantially more variants tested with only a modest reduction in mappability rate (Fig. 1h). Full coverage of DHSs was limited primarily by sequencing depth, suggesting that additional sequencing would yield additional power. The imbalance was less frequent at highly accessible DHSs (Supplementary Fig. 4, Supplementary Fig. 5), consistent with our previous observations of buffering of point variants at strong sites[4,5].

In the F1 offspring of an inbred cross, each variant on a given chromosome is in perfect linkage. Thus we considered the power of our approach to detect focal alteration of individual DHSs rather than coordinately altered chromatin accessibility. By examining the co-occurrence of imbalance of nearby variants, we found that allelic ratios of nearby sites were strongly correlated only at distances less than 250 bp, well below the median width of a DHS hotspot (Fig. 1i). This suggests that our approach offers high resolution to identify sequence variation leading to local effects on chromatin state.

**Cellular context-sensitivity**. We assessed the cell-type accessibility patterns in 39 diverse cell and tissue types by the ENCODE project, all mapped in reference C57BL/6 mice[24], and excluding liver, lung, kidney, or B cells. We categorized SNVs based on whether accessibility was higher at the reference (C56BL/6J) or the non-reference allele (Fig. 2a). Both sets of imbalanced SNVs showed increased cell-type selectivity with respect to SNVs not affecting accessibility. But nearly half of the non-reference higher sites had evidence for a DHS in another cell or tissue type in C56BL/6J mice, a 3-fold enrichment compared to a background set of mappable SNVs in inaccessible DNA and thus not tested

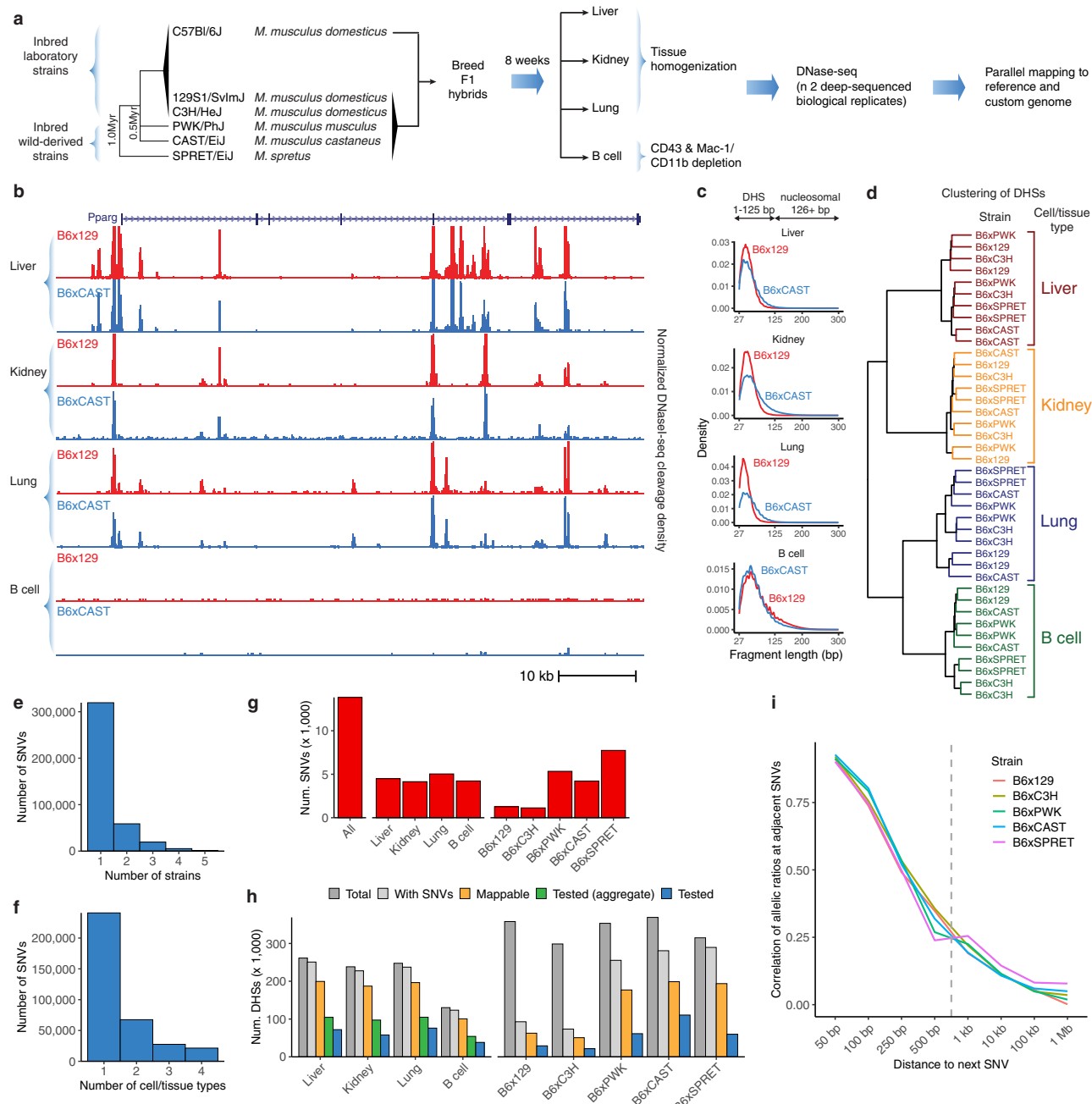

**Fig. 1 Allelic analysis of DNA accessibility in hybrid mice from diverged strains. a** Overall schematic of experiment **b**. DNase-seq profiles at the Pparg locus in liver, kidney, lung tissue, and B cells from F1 crosses of C57Bl/6J dams with 129/SvImJ and CAST/EiJ sires. **c** Fragment length distribution of samples showing high-quality libraries comprising non-nucleosomal fragments. **d** Hierarchical clustering of DHSs from high-depth samples. **e–f** Counts of SNVs shared across strains (**e**) and cell types (**f**). **g** Counts of imbalanced SNVs (FDR 10%). Counts are reported in aggregate across all data sets (left), by cell type (middle), and by parental strain (right). **h** Summary of the master list of DHSs overlapping SNVs from all strains. Counts include all DHSs (dark gray), DHSs with SNVs (light gray), DHSs passing mappability filters (orange), DHSs with sufficient coverage to test for imbalance across all data sets (green) and in individual cell types or strains (blue). Counts include only autosomal DHSs. **i** Pearson correlation of allelic ratios at adjacent SNVs broken down by distance to next SNV. The dashed line represents the median width of DHS hotspots overlapping SNVs in this study.

for imbalance (see "Methods" section). This suggests that point changes affecting accessibility at sites with preexisting activity act more frequently by broadening DNA accessibility to other cell and tissue types, rather than de novo evolution of novel regulatory DNA. Only a minority of non-reference higher sites overlapped a DHS in a cognate cell or tissue type, suggesting the majority were qualitative creation rather than quantitatively increased accessibility (Fig. 2b). This cell-type-specific expansion

of accessibility drew broadly from other cell lineages with an only moderate preference for related cell types (Fig. 2b).

We then examined the cell-type selectivity of imbalance itself. We were able to test for imbalance per cell type (combining data from different strains) at an average of 196,276 SNVs per cell type (Table 1). We identified clear examples of strong imbalance across multiple strains specific to a particular cell type (Fig. 3a, b). In both examples, cell-type-specific imbalance in one DHS was

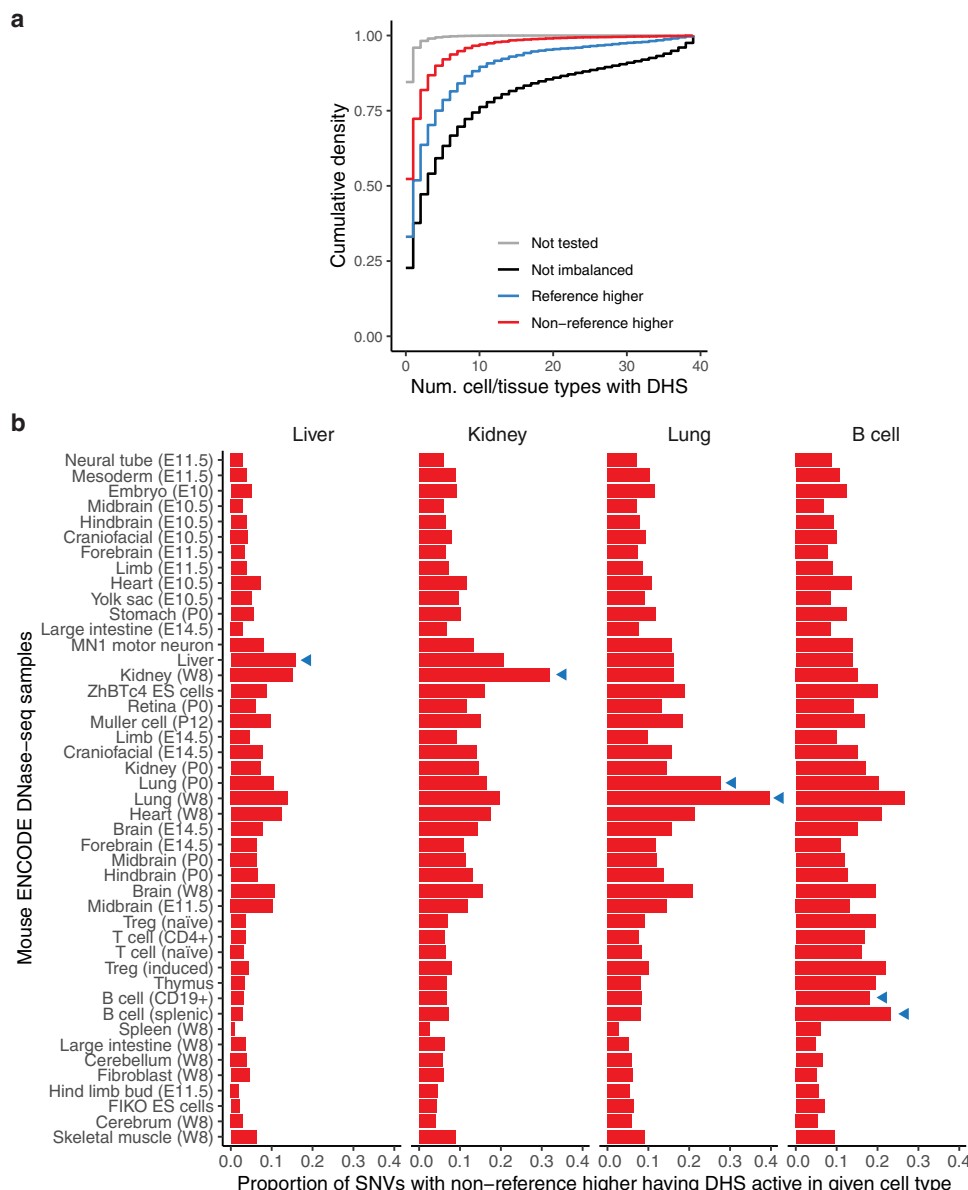

**Fig. 2 Predetermination of sites of strain-specific DNA accessibility. a** Cumulative density distribution of cell-type activity of DHSs measured across 39 mice ENCODE DNase-seq samples in reference C57BL/6 mice[24]. DHSs are stratified based on whether imbalance favored the reference or non-reference allele. Not tested refers to the set of mappable SNVs not in DHSs for Liver, Kidney, Lung, or B cells and therefore not tested for imbalance. **b** The proportion of imbalanced SNVs from a given cell/tissue type that overlap DHSs from mouse ENCODE cell and tissue types (along the y-axis). Cell and tissue types are ordered based on the hierarchical clustering of all hotspots. Developmental time points for some samples are indicated in parenthesis (E, embryonic day; P, postpartum, W; adult week). Blue triangles indicate ENCODE samples matching tissues from hybrid mice in this study.

### Table 1 SNVs tested for imbalance per cell type/strain.

| Strain/cell type | Liver | Kidney | Lung | B cell | All cells/tissues |
|---|---|---|---|---|---|
| B6x129 | 28,527 | 11,353 | 11,262 | 11,423 | 52,400 |
| B6xC3H | 22,526 | 12,423 | 3932 | 4481 | 37,915 |
| B6xCAST | 78,740 | 37,576 | 92,128 | 45,777 | 215,629 |
| B6xPWK | 37,819 | 34,325 | 23,285 | 5,880 | 103,441 |
| B6xSPRET | 45,858 | 11,100 | 16,995 | 29,439 | 113,398 |
| All hybrids | 187,307 | 110,643 | 151,818 | 94,469 | 357,303 |
| Imbalanced | 4490 | 4147 | 5037 | 4230 | 13,835 |

Shown are counts for variants tested for imbalance in per-sample, per-cell type, and per-strain analyses. Bottom row shows imbalanced variants for the per-cell type analysis.

associated with a coordinate change in accessibility at a nearby DHS (Fig. 3a, b), though we note that it is not possible to infer the direction of causality. Overall, however, we identified a higher degree of sharing of imbalance between samples of the same cell type than from the same strain or unrelated samples (Fig. 3c). Pairwise comparison of different cell types showed an average of 63% sharing of imbalanced sites $(1 - \pi_0)$, suggesting a high prevalence of genetic effects demonstrating cell-type context sensitivity (Fig. 3d).

**TF-centric analysis of variation.** We then asked to what extent variation affecting DNA accessibility in *cis* was linked to direct perturbation of TF recognition sequences. We scanned the mouse reference and strain-specific genomes using motif models for

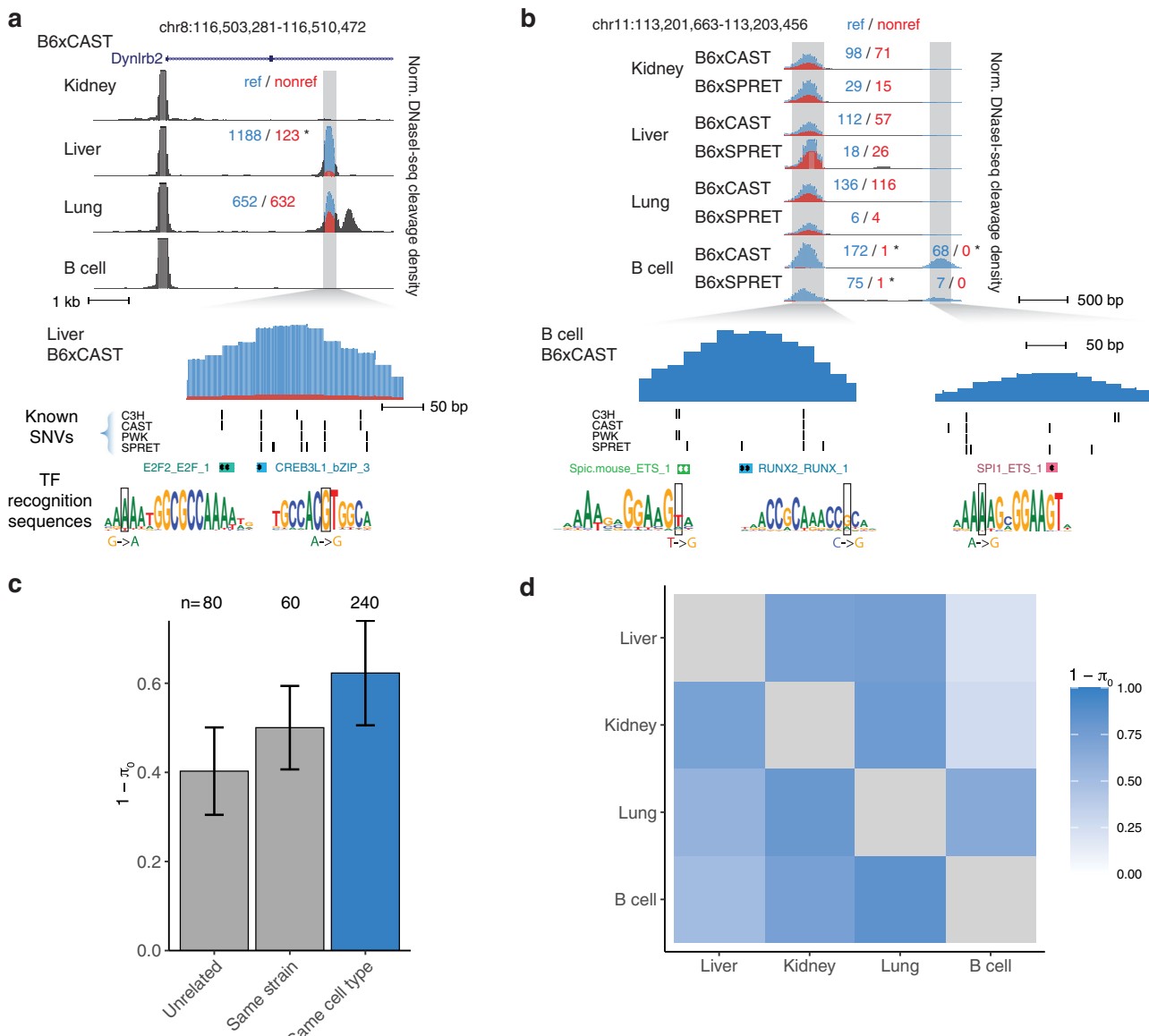

**Fig. 3 Cross-cell type analysis of allelic variation in DNA accessibility. a, b** Example DHSs showing cell-type-specific imbalance. Normalized DNaseI cleavage density is colored by signal mapping to reference (blue) and non-reference (red) alleles based on the aggregation of informative SNVs. Counts by peaks denote the sum of reads mapping to reference and non-reference alleles for all SNVs in region; *indicates statistically significant allelic imbalance. Selected TF recognition sequences overlapping imbalanced SNVs are highlighted below. **a** DHS accessible in liver and lung but with imbalance only in the liver. **b** DHSs accessible in all 4 tissues (left) or specific to B cells (right); both DHSs only show an imbalance in B6xCAST and B6xSPRET B cells. **c–d** Sharing of imbalance by cell type. $1 - \pi_0$ represents the proportion of rejected null hypotheses by Storey's method. **c** Average sharing of imbalance ($1 - \pi_0$) between samples of the same cell type vs. samples sharing only the same strain or unrelated samples (sharing neither strain nor cell type). Bar height represents the mean of all pairwise comparisons; error bars represent standard deviation. **d**. Pairwise sharing of imbalance between cell or tissue types.

2203 TFs[4]. We found that while only a small fraction of imbalanced variation overlapped a recognition sequence for any individual TF, 61% of variation overlapped stringent motif matches (FIMO $P < 10^{-5}$) when considering all TFs with known motifs (Fig. 4a). Imbalanced SNVs were found more frequently at sites of DNase I footprints, contingent on the presence of a recognizable TF recognition sequence (Fig. 4b). We found that aggregate imbalance was concentrated over the core positions of the motif for many key TFs (Fig. 4c). Sensitivity profiles for human TFs generated using previously published allelic accessibility data[4] largely resembled those generated from mouse data, although some factors such as HNF1A showed significant enrichment only in the mouse data (Fig. 4d).

We next performed an analysis of cell-type-specific imbalance calls at TF recognition sequences. We found higher rates of cell-type-specific imbalance at sites of DNase I footprints in matching cell and tissue types, relative to unmatched cell and tissue types (Fig. 5a). We found that distinct TF families presented varying cell-type-specific patterns of enrichment of imbalanced SNVs over their motifs (Fig. 5b). For example, JDP2 (AP-1) only showed enrichment in the lung (Fig. 5c), and ETS factors showed the highest enrichment in B cells (Fig. 5d). In both cases, no enrichment is evident when data are aggregated across multiple cell and tissue types. Other factors showed patterns of enrichment across a subset of cell types: HNF factors showed peak enrichment in liver and kidney (Fig. 5e), while CEBP showed

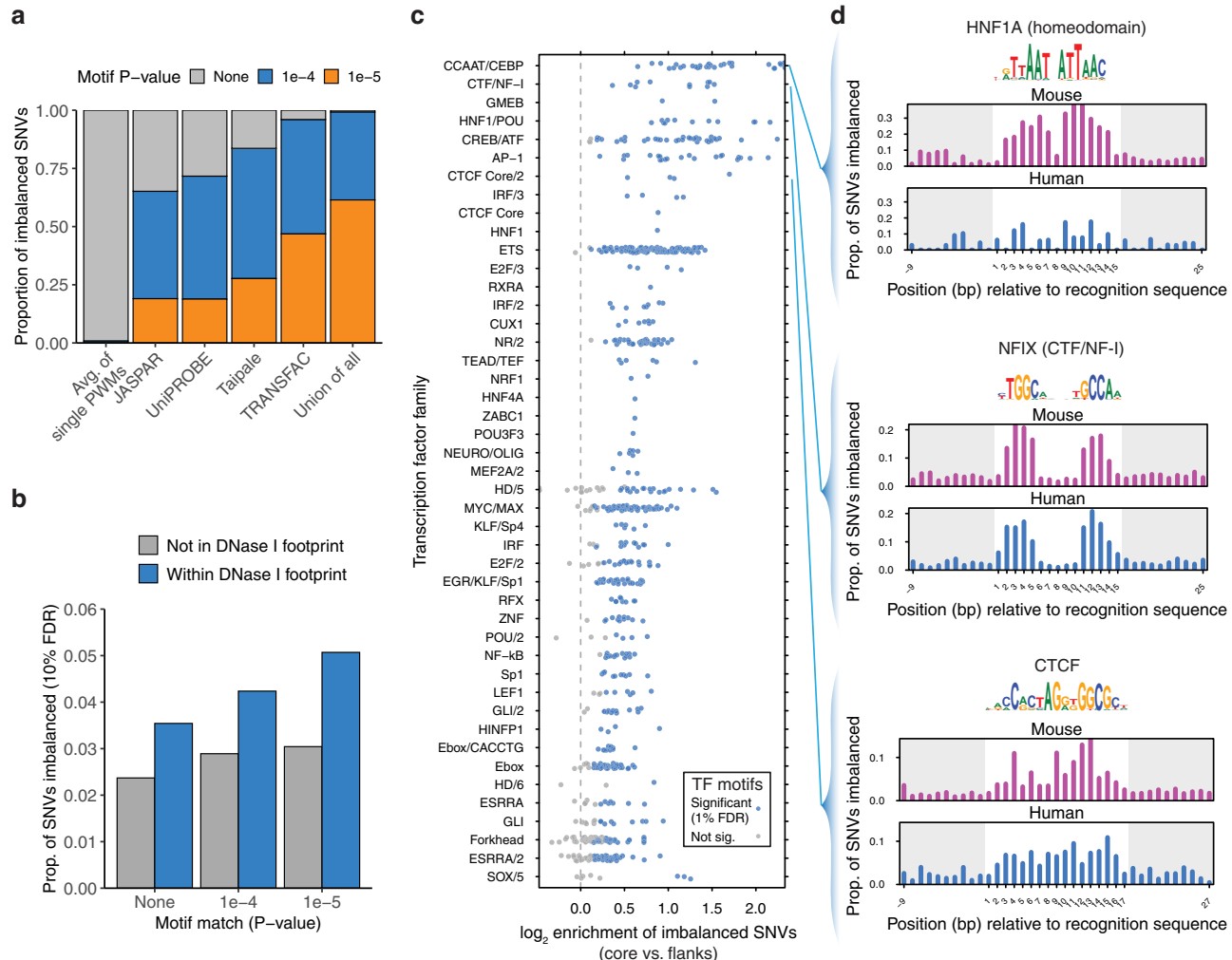

**Fig. 4 Analysis of variation affecting TF occupancy. a** Overlap of imbalanced SNVs with matches to TF motifs from different large-scale collections. **b**. Frequency of aggregate imbalance at SNVs overlapping TF motifs from a large-scale SELEX-seq database[48] and DNase I footprints aggregated across all cell types. Results are stratified by the FIMO value of overlapping TF motifs (if any). $n = 357,303$ SNVs. **c** Enrichment by TF family of imbalanced SNVs in TF core recognition sequences, relative to flanking sequence. Each point corresponds to a TF motif, grouped into TF families on the Y-axis. Shown are TF families with at least one enriched motif. **d** TF profiles for NFIX, CTCF, and HNF1A. For comparison, profiles generated from published analysis in human[4] are shown below in blue.

enrichment in lung and liver (Fig. 5f). These results suggest that cell-type-specific identification of imbalanced variants can yield a more accurate assessment of variants affecting TF occupancy than aggregate analyses across multiple cell types.

To facilitate recognition of sequence variation affecting DNA accessibility in the mouse and human genomes, we incorporated the mouse data into our Contextual Analysis of Transcription Factor Occupancy (CATO2) scoring approach[4]. CATO2 trains a logistic regression model for each TF motif on a variety of genomic annotations and TF-centric parameters. By standardizing genomic annotations between humans and mice, we directly incorporated both data sets (Fig. 6a). Combining the mouse and human data yielded a dramatic increase in TF families with sufficient variation (Supplementary Table 3). In addition to the inherent cell-type selectivity of DHS tracks, we incorporated per-cell type imbalance data in two ways (see "Methods" section): (i) TF models were trained on the subset of mouse cell types demonstrating enrichment of imbalanced SNVs over the recognition sequence (Fig. 6b); and (ii) a sparse generalized linear model was trained to establish cell-type specific weights for the contribution of each TF model to the overall score (Fig. 6c).

Assessing performance on a pair of DNase-seq datasets generated in B6xCAST mouse embryonic stem cells (mESCs) (Supplementary Table 4) showed that CATO2 retained performance even on a completely independent validation set (Supplementary Fig. 6). Furthermore, assessment of predictive performance for CTCF directly against matching ChIP-seq data showed that CATO2 scores were also predictive for allelic TF occupancy (Supplementary Fig. 6). In addition, cell-type-specific models showed increased predictive performance using precision-recall analysis (Fig. 6d, Supplementary Fig. 7). These results suggest that CATO2 provides a strong foundation for the assessment of functional non-coding variation.

**Allelic effects on transcript levels.** The activity of distal regulatory elements is compartmentalized and shows highly specific interactions with certain genes[25]. To examine the effect of altered accessibility on steady-state transcript levels, we performed RNA-seq in a subset of matching samples (Supplementary Table 5). We analyzed allelic expression measured by RNA-seq using a similar pipeline to that used for the DNase-seq data (Methods). We then

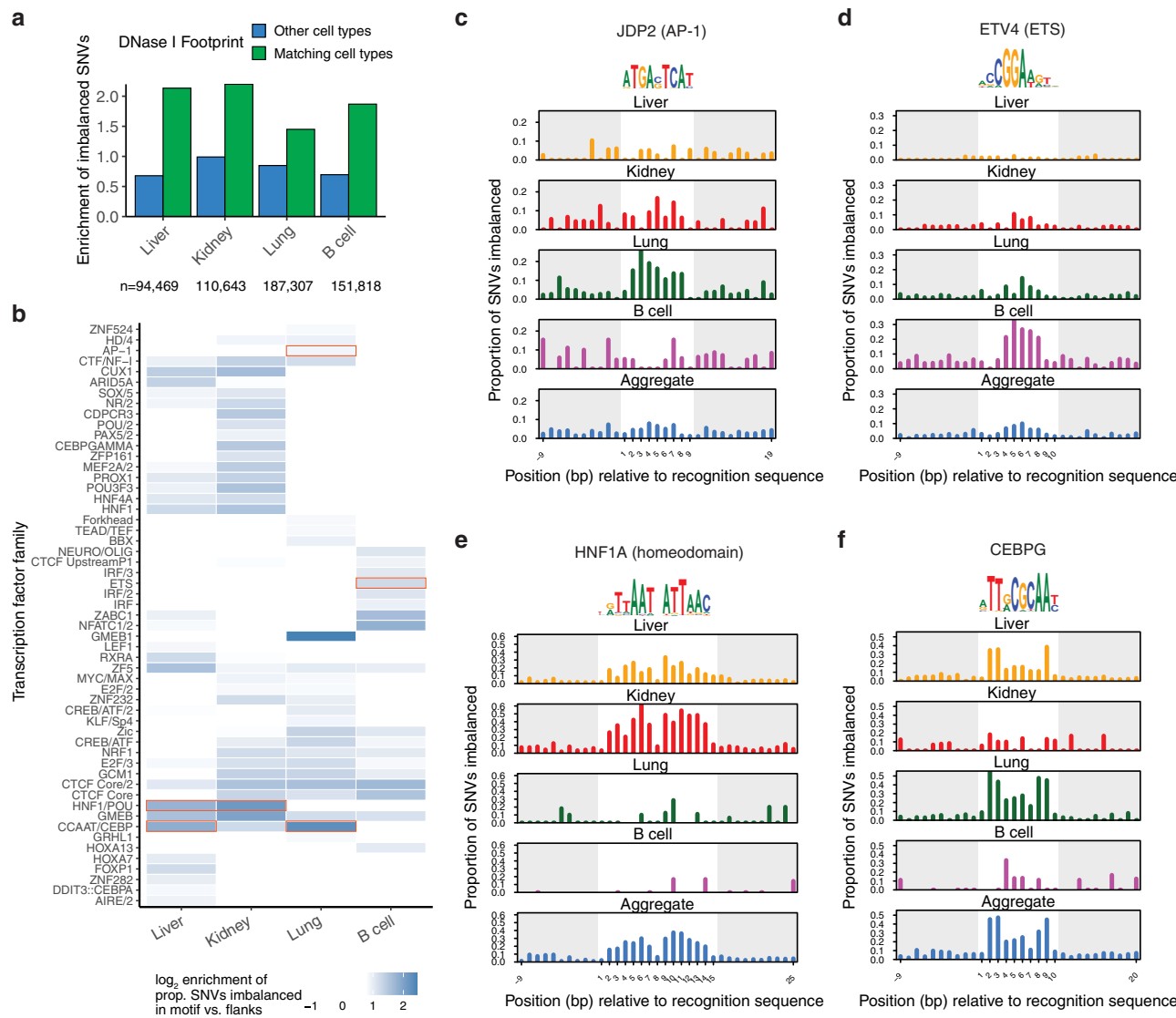

**Fig. 5 Cellular context-sensitive analysis of variation affecting TF occupancy. a** Enrichment of imbalance called in each cell type for overlap with DNase I footprints in matching cell type (green) or in other cell types (blue). **b** Cell-type-specific enrichment of SNVs in motif for TFs. Shown are TF families with greater than twofold enrichment in at least one cell type. **c**–**f** Analysis of variation affecting TF occupancy across cell/tissue types for JDP2, ETV4, HNF1A, and CEBPG motifs.

compared allelic accessibility at DHSs to allelic transcript levels linked to transcription start sites (TSS) within 500 kb (Fig. 7). We detected correlation significantly above that observed in permuted data extending distances as far as 100 kb surrounding the TSS. Maximal correlation (*R* between 0.1 and 0.2) occurred within 10 kb of the TSS and was slightly higher downstream than upstream. Our work suggests that long-range regulatory interactions between distal accessible sites and genes are common genome-wide and are amenable to analyses using the resources and approach we have described herein.

## Discussion

Our work shows that most *cis*-linked differences in DNA accessibility among diverged mouse genomes can be attributed to the direct perturbation of TF recognition sites. Past reports have differed on the degree of allelic occupancy that can be linked to point changes in TF recognition sequences, ranging from 9% for NF-kB[14] to 85% for CTCF[5]. Yet, studies of a single TF are confounded by the possibility that changes in its recognition

sequence may perturb the binding of other factors, either at the same site or a nearby one. By analyzing a broad set of TFs with known sequence specificities, we identify that fully 61% of imbalanced sites can be linked to changes in TF recognition sequences (Fig. 4a). We expect that the range of enrichment of imbalanced SNVs in TF motifs observed in Fig. 4b reflects both the role of cooperative binding and the accuracy of binding site recognition for individual TFs. Given the challenge of obtaining TF-specific occupancy data for all factors expressed in a given cell type, we expect that improved recognition of in vivo occupied TF binding sites from DNase I footprinting data[26,27] will be the most fruitful way to obtain further improvements in prediction performance.

Given that only a select minority of SNVs affect TF binding in a given cell type, additional large-scale analyses are needed to functionally assess noncoding variation in context. Our work shows that highly diverged mouse subspecies (including CAST/EiJ, PWK/PhJ, SPRET/EiJ) provide an efficient system for assessing regulatory variation that overcomes the low density of polymorphism in human populations. Compared to past work in human[4], the present

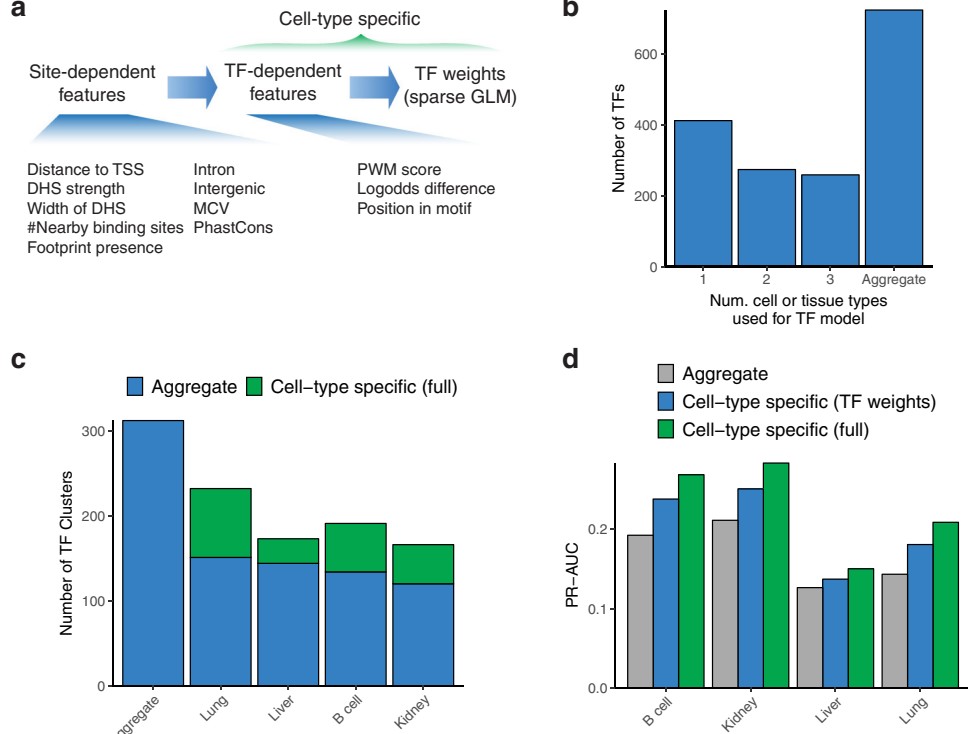

**Fig. 6 Cell-type-specific prediction of variation affecting TF occupancy. a.** CATO2 strategy for cell-type-specific scoring of regulatory variation. **b**. The number of mouse cell types used for each TF model; all TF models included human data. **c**. The number of unique TF clusters with non-zero coefficients in aggregate and cell-type-specific CATO2 scores. TFs shared with the aggregate model are highlighted in blue. **d** The area under precision-recall curves (full curves are shown in Supplementary Fig. 7) showing performance to predict imbalanced polymorphism on SNVs tested for imbalance in individual cell types.

work required only 14% of the samples and half the sequencing depth, yet it yielded two orders of magnitude more SNVs tested for cell-type-specific imbalance (avg. = 136,059 SNVs per cell type). This power enables cell-type specific analyses that uncover context-sensitive variation otherwise masked by aggregation of data across multiple cells or tissue types. The use of mice also enables ready access to a variety of cell and tissue types difficult to access in humans[21,28]. The high rate of imbalance in cell-type-specific DHSs underscores the importance of robust sequencing depth across a full spectrum of cell types and suggests that efficient generation of additional profiling data in other cell and tissue types from these strains will efficiently increase the power of TF-centric models to recognize functional variation.

Cross-species TF-centric analysis of genomic variation overcomes the low sequence conservation of the *cis*-regulatory landscape[24] by obviating the need for direct analysis of human regulatory variants at the mouse locus and enables scalable prediction of previously unseen variation. While CATO2 presently requires cell-type specific variation data to train TF weights, inference of TF weights from other more readily available information, such as measurements of TF expression and activity, may also be possible. Such an approach could enable the classification of functional regulatory variants in cell and tissue states without directly measured genetic data. Supporting this possibility, nearly half of strain-specific imbalance represented an expansion of accessibility at known DHSs to a new cell or tissue type. We speculate that functional regulatory variation might most easily arise from the creation of TF recognition sequences that expand the selectivity of an existing DHS by acting cooperatively with existing TF recognition sequences. It would also be straightforward to incorporate *trans*-regulatory differences between strains or species into future models to enable an analysis of *trans*-regulatory effects on gene expression[29–31].

The global correlation observed between allelic accessibility and allelic transcript levels were statistically significant but modest. Much as the majority of point variants are buffered in terms of their effect on local chromatin features[5], enhancer networks controlling gene expression likely demonstrate a high degree of redundancy and selectivity[25,32,33]. The correlation we observe could serve as a benchmark for the development of genome-wide methods to predict likely target genes of distal regulatory elements and complements systematic locus-scale investigation of regulatory architecture using genome engineering[33,34]. Thus it is likely that further exploitation of mouse genetics will provide the substrate for more granular models of enhancer-promoter interaction.

## Methods

**Mouse husbandry.** The mice used in this study were F1 hybrids of C57Bl/6J reference females with wild-derived strains 129/SvImJ (B6x129), C3H/HeJ (B6xC3H), CAST/EiJ, (B6xCAST), PWK/PhJ (B6xPWK), and SPRET/EiJ (B6xSPRET). 129/SvImJ and C3H/HeJ hybrid females were acquired from the Jackson Laboratory (8 weeks old, housed 4/cage). CAST/EiJ, PWK/PhJ, SPRET/EiJ inbred males were acquired from the Jackson Laboratory and were bred to C57Bl/6J female mice at FHCRC. Mice were maintained on a 12-h light, 12-h dark schedule with lights turned on at 7 a.m. The housing room was maintained at 20–24 °C with 30–70% relative humidity. Mice have housed in individually ventilated cages (Allentown) with 75 square inches of floor space and 60 air changes/hour. Biofresh cage bedding was (Absortion Corp) at 1/8 inch depth and autoclaved on site. Water and Purina 5053 (irradiated) were available ad libitum. Nestlet material (Envigo's diamond twist 7979C, also irradiated) was present in each cage for enrichment. Autoclavable certified igloos (Bio-serv) were provided in some cages. Mice were housed in a barrier facility that is AAALAC accredited. Mice were sacrificed at 8 weeks of age by $CO_2$ asphyxiation. All work was approved by the Institutional Animal Care and Use Committee (IACUC) of the Fred Hutchinson Cancer Research Center (FHCRC).

**Nuclei isolation from mouse tissues.** Solid mouse tissues were typically obtained from 4 mice sacrificed together with their tissues pooled. The whole liver (all lobes),

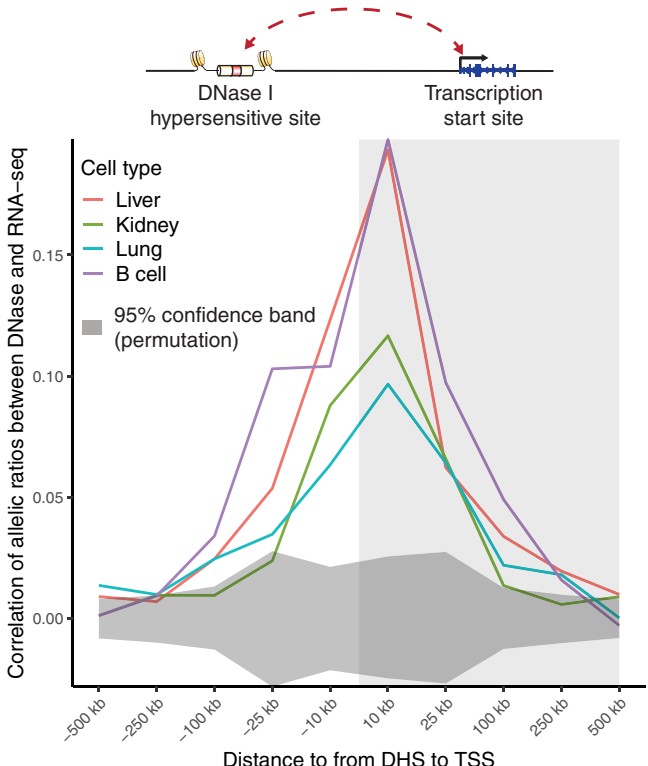

**Fig. 7 Imbalanced accessibility and transcript levels.** Pearson correlation in allelic ratios between SNVs in DHSs and transcript levels are broken down by distance to the transcription start site (TSS). All pairs of DHSs and TSSs within 500 kb are considered. Dark gray shading at the bottom indicates a 95% confidence band from 1000 permutations of DHS allelic ratios among DHS-TSS pairs for each cell or tissue type. Light gray shading at right indicates that DHS lies downstream of TSS.

both kidneys, and all lobes of the lungs were rapidly dissected. Tissues were minced in 2 mm square pieces and resuspended in 5 mL of homogenization buffer (20 mM tricine, 25 mM D-sucrose, 15 mM NaCl, 60 mM KCl, 2 mM MgCl$_2$, 0.5 mM spermidine, pH 7.8) per tissue. Nuclei were released using 5–10 strokes in a Dounce homogenizer with a loose-fitting type-A pestle and the resulting homogenate was filtered through a 120 µm filter. Samples were returned to the Dounce for 5–10 strokes with a tight-fitting type-B pestle, and filtered using a 40 µm mesh filter. 5 mL of homogenate was mixed with 3 mL of 50% Optiprep solution and layered onto a 4 mL 25% −1 mL 35% two-step Optiprep gradient and centrifuged for 20 min at 6100×$g$ in a swinging bucket rotor. The nuclei pellet was washed once in 10 mL of buffer A (15 mM Tris-HCl, 15 mM NaCl, 60 mM KCl, 1 mM EDTA, 0.5 mM EGTA, 0.5 mM spermidine) and resuspended at a concentration of 2 × 10$^6$ per mL.

Marrow was obtained from femurs of 8-week-old female mice. B cells were isolated using an AutoMACS (Miltenyi Biotech) to deplete CD43 and Mac-1/CD11b markers. Cells were washed once with Dulbecco's PBS (without MgCl$_2$ or CaCl$_2$). Nuclei were extracted by resuspending cells in buffer A supplemented with 0.015% detergent (IGEPAL-CA630) (Sigma) and incubating for 5–10 min on ice. Following incubation, the nuclei were collected by centrifugation (600×$g$) and resuspended in buffer A at a concentration of 2 × 10$^6$ nuclei per mL.

**DNase I digestion of mouse nuclei.** Fresh nuclei were incubated for 3 min at 37 °C with limiting concentrations of the DNA endonuclease deoxyribonuclease I (DNase I) (Sigma) in buffer A supplemented with Ca$^{2+}$. The digestion was stopped with 5× stop buffer (125 mM Tris-HCl, 250 mM NaCl, 0.25% SDS, 250 mM EDTA, 1 mM spermidine, 0.3 spermine, pH 8.0) and the samples were treated with proteinase K and RNase A. The small 'double-hit' fragments (<250 bp) were recovered by sucrose ultracentrifugation, end-repaired, and ligated with adapters compatible with the Illumina sequencing platform. Libraries were amplified using minimal PCR cycles based on a trial qPCR amplification (8–16 cycles) (Supplementary Table 6). A detailed protocol describing the genome-wide mapping of DNase I hypersensitivity can found in[35].

**mESC profiling.** B6xCAST mESCs were cultured on plates coated with 0.1% gelatin (EMD Millipore ES-006-B) in 80/20 medium comprising 80% 2i medium and 20% mESC medium. 2i medium contained a 1:1 mixture of Advanced DMEM/

F12 (ThermoFisher 12634010) and Neurobasal-A (ThermoFisher 10888022) supplemented with 1% N2 Supplement (ThermoFisher 17502048), 2% B27 Supplement (ThermoFisher 17504044), 1% Glutamax (ThermoFisher 35050061), 1% Pen-Strep (ThermoFisher 15140122), 0.1 mM 2-Mercaptoethanol (Sigma M3148), 1250 U/ml LIF (ESGRO ESG1107l), 3 µM CHIR99021 (R&D Systems 4423) and 1 µM PD0325901 (Sigma PZ0162). mESC medium contained Knockout DMEM (ThermoFisher 10829018) supplemented with 15% Fetal Bovine Serum (FBS, Bench-Mark 100–106), 0.1 mM 2-Mercaptoethanol, 1% Glutamax, 1% MEM Non-Essential Amino Acids (ThermoFisher 11140050), 1% Nucleosides (EMD Millipore ES-008-D), 1% Pen-Strep and 1250 U/ml LIF. HEK-293T cells were cultured in DMEM supplemented with 10% FBS, 1 mM sodium pyruvate (ThermoFisher 11360070), 1% Glutamax, and 1% Pen-strep. All cells were grown at 37 °C in a humidified atmosphere of 5% CO2 and passaged on average twice per week. For ChIP-seq, mESCs were crosslinked for 10 min in 1% formaldehyde and quenched in 125 mM glycine. Four hundred microgram of chromatin was sheared by Covaris LE220 Ultrasonicator (Covaris). Eighty microliter of CTCF antibody (Cell Signaling 2899S) was conjugated to M-280 Dynabeads (Invitrogen 11204D) for 6 h at 4 °C, followed by overnight immunoprecipitation. After reversing crosslinks, immunoprecipitated DNA was treated with Proteinase K and RNase A and purified using the DNA Clean and Concentrate-5 Kit (Zymo Research).

**Short-read sequencing and processing.** DNase-seq and ChIP-seq libraries were sequenced on an Illumina HiSeq 2500 by the High-Throughput Genomics Center (University of Washington) or a NextSeq 500 (NYU Institute for Systems Genetics) in paired-end 36 bp mode.

Short reads were first trimmed to remove low-quality sequence or adapter contamination using trimmomatic v0.33[36] with parameters 'TOPHRED33 ILLUMINACLIP: TruSeq3-PE-2.fa:2:5:5:1:true MAXINFO:27:0.95 TRAILING:20 MINLEN:27'.

To reduce potential reference mapping bias, custom strain-specific genomes were created using vcf2diploid v0.26a[37] to incorporate known[22] point variants and insertions or deletions (REL-1505-SNPs_Indels / version 5). Chain files were created for use with the UCSC liftOver tool to enable the genomic coordinate conversion between the reference and strain-specific genomes. Genomes included unscaffolded contigs and alternate sequences but not the Y chromosome.

Reads were mapped using Burrows-Wheeler Aligner (BWA) v0.7.13 to both the mouse reference assembly (GRCm38/mm10) and the appropriate strain-specific genome with the command 'bwa aln -n 0.04 -l 32 -t 2 -Y[38]. Alignments were post-processed with a custom Python script using pysam [https://github.com/pysam-developers/pysam] to retain only properly-paired or single-end reads mapping uniquely to the autosomes and chrX with a mapping quality of at least 20. Paired-end reads were required to have an inferred template length of less than 500 bp. Duplicate reads were flagged on a per-library basis using SAMBLASTER v0.1.22[39]. Mapped tags were converted to BED format using awk and bedops v2.4.35[40]. DNase I hypersensitive sites were identified using hotspot2 v2.1.1[41]. Reference mm10 coordinates were used for all analyses except for read counting (which additionally relied on the strain-specific mappings).

**Assessment of allelic imbalance.** Reads overlapping all known point variants were assessed for allelic imbalance at all SNVs overlapping a DNase hotspot (5% FDR) called on the aggregate of all DNase data for a given strain and cell or tissue type. Reads were extracted from DNase-seq alignments using a custom script countReads.py written in Python and pysam. The liftOver tool was used with the chain file generated by vcf2diploid to map variant coordinates from mm10 to each strain-specific genome. Reads were required to map uniquely to both mm10 and the strain-specific reference with the same mapping quality and template length. We excluded 3 bp at the 5′ end of the read to exclude any possibility of sequence-specific DNase I cut rate[42]. Only reads with a base quality >20 at the variant position were counted. Read pairs overlapping a variant were counted once. 2 additional mismatches were permitted besides the known variant. Duplicate reads passing all filters with the same 5′ position on the reference were excluded (independent of the SAM duplicate flag). Variants lying within 72 bp of a known insertion or deletion or with ≤60% of total overlapping reads passing filters were excluded from further analysis.

To minimize possible mapping bias, we generated a mappability track by mapping simulated 36-bp paired-end reads with up to 125 bp-fragment length overlapping known SNPs and including no sequencing errors. Simulated reads were mapped back to both the reference and strain-specific genomes and filtered using the approach described above. SNVs having ≤95% of simulated reads mappable were filtered out.

A background set of SNVs not tested for imbalance was identified as all mappable SNVs not overlapping a DHS in the master list or any individual condition.

Allele counts from all samples were aggregated into a single matrix and analyzed separately for per-sample, per-strain, and per-cell type imbalance. Only SNVs with at least 30 reads in one condition were retained. To account for variable sequencing depth and enrichment, we fit a beta-binomial distribution for each condition using sites with >100 reads and computed $P$ values against an expected 50% of reads mapping to each allele. We accounted for multiple testing using a false discovery rate (FDR) cutoff of 10% using the R package $q$-value v2.14.1[43]. Aggregate imbalance analyses used sums of per-cell type counts.

**Transcription factor motif analysis**. We scanned the reference and all strain-specific genomes using FIMO v4.10.2[44] with TF motifs and TF clusters as in[4]. Strain-specific motif matches were converted to mm10 coordinates using liftOver, and a non-redundant list of motif matches per strain was created from the union of both sets.

We analyzed the intersection of SNVs tested for imbalance with these motifs. We considered motifs with a median of ≥40 SNPs per position in the motif and ≥3 positions with ≥7 significant SNPs; positions with <7 SNPs were considered missing data. For SNPs overlapping multiple matches to the same motif, we chose the best motif instance per SNP on the basis of the FIMO P value.

**Genomic annotation**. SNVs were annotated accordingly:

- Cell-type activity spectrum MCV (multi-cell verified) was computed from a set of 45 representative samples from Mouse ENCODE selected through hierarchical clustering analysis. A master list[45] was generated from these samples and MCV was scaled from 0 to 1 by dividing by 45.
- Footprints on the mouse and human samples were called using FTD[27].
- RefSeq Genes and CpG Islands were downloaded from the UCSC Genome Browser.

Human SNPs were annotated as in ref. [4]. Quantitative mouse annotations were scaled by the ratio of the mean annotation value at SNPs in mouse vs. human. Parameters were standardized to have a mean of 0 and a standard deviation of 1.

**CATO2 scores**. We generated CATO2 models on the combined human and mouse data as in ref. [4] with several modifications. First, we trained a logistic model for the genomic annotations at each SNV using the glm() function in R v3.5.2:

```
significant ~ MCV^2 + intron + intergenic + log(Dist. to
TSS)^2 + DHS strength^2 + log(Width of DHS) + Footprint
presence + #nearby binding sites^2 + PhastCons
```

Then, we trained a second glm() logistic model for each TF, which incorporated the global per-SNV score as a parameter. Imbalance was analyzed per-cell type for the mouse data and cell types demonstrating log enrichment >1 of imbalanced SNVs over the recognition sequence.

```
significant ~ global.fit + log(score)^2 + logodds
difference + x_2 + ... + x_n
```

Finally, we combined scores from individual TF models at each SNV using the R package GLMnet v2.0-16[46] to train a sparse generalized linear model (GLM) using the lasso penalty and 50-fold cross-validation with performance measured by AUC. To score human point variants, annotation values were computed and standardized as before and CATO2 scores were computed using the R function predict(type = "response").

**Generation and analysis of RNA-seq data**. Total RNA was isolated using the mirVana miRNA Isolation Kit with phenol (AM1560). Spike-in controls were mixed in (Ambion-ERCC Mix, Cat no. 4456740) and Illumina sequencing libraries were made using the RNA TruSeq Stranded total RNA (Illumina). Libraries were sequenced on an Illumina HiSeq 2500 or NextSeq by the High-Throughput Genomics Center (University of Washington) in paired-end 36 bp or 76 bp modes. Previously published data for kidney, liver, and lung B6xCAST[19] were downloaded from the NCBI SRA repository (Supplementary Table 5).

Reads were mapped to the mm10 reference and strain-specific genomes in parallel using STAR v2.5.2a[47]. Counts from all non-exonic SNVs overlapping a given Gencode M10 basic level 1 and 2 protein-coding transcripts were aggregated. SNVs were analyzed using the same allele counting pipeline as for DNase-seq data. We assessed allelic imbalance using a beta-binomial model fit at SNVs with >100 reads. We accounted for multiple testing using a false discovery rate (FDR) cutoff of 10% using the R package qvalue[43] and additionally required >60% of reads to map to one allele. Counts were aggregated for all samples per cell type and per-DHS hotspot. A minimum of 50 total reads per transcript was required. RNA-seq imbalance data were then overlapped with per-sample DHS imbalance data.

**Reporting summary**. Further information on research design is available in the Nature Research Reporting Summary linked to this article.

## Data availability

The data that support this study are available from the corresponding author upon reasonable request. All sequencing data generated for this study have been deposited in the NCBI GEO repository under accession GSE156692. RNA-seq data for B6xCAST tissues were downloaded from SRA study SRP020526. DNase-seq data were downloaded from the Mouse ENCODE project under accessions ENCLB033TJA, ENCLB055DAX, ENCLB087FMN, ENCLB123ELX, ENCLB132WYL, ENCLB144OVE, ENCLB144XPP, ENCLB255IYN, ENCLB268SEP, ENCLB271XFH, ENCLB309AYR, ENCLB310LJW, ENCLB329BXZ, ENCLB330JBO, ENCLB338AZH, ENCLB369NIA, ENCLB388UBV, ENCLB414QZO, ENCLB439JPP, ENCLB475GJF, ENCLB475XWE, ENCLB491WIB, ENCLB509ASX, ENCLB519UHH, ENCLB523BNO, ENCLB527TCS, ENCLB551IVD, ENCLB558NIW, ENCLB568ONV, ENCLB584XBJ, ENCLB588UWH, ENCLB667FFY,

ENCLB693PQW, ENCLB697VFP, ENCLB699GWH, ENCLB709OVK, ENCLB788JUN, ENCLB792YOE, ENCLB792ZMJ, ENCLB854PNT, ENCLB893AUS, ENCLB921AZQ, ENCLB933JCH, ENCLB939IXH, and ENCLB995XTX. The source data are provided with this paper.

## Code availability

The processing pipelines for DNase-seq and RNA-seq data are available on Github [https://github.com/mauranolab/hybridmouse]. All code for analyses herein is available upon request.

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

## Acknowledgements
We thank Daniel Bates, Morgan Diegel, Douglass Dunn and Fidencio Neri (Altius Institute) for assistance with library construction and sequencing. We thank Eric Rynes (Altius Institute) and Nick Vulpescu (NYU Institute for Systems Genetics) for technical assistance. This work was partially funded by NIH Grants U54HG007010 and 1S10OD017999 to J.A.S. and R35GM119703 (M.T.M).

## Author contributions
R.B., J.M.H., M.S.H., R.O., and M.T.M. performed experiments. M.A.B. and M.G. supervised mouse work. M.T.M. analyzed data. M.T.M. and J.A.S. wrote the manuscript.

## Competing interests
The authors declare no competing interests.
