## [Peer Review File · Nature Communications]

REVIEWER COMMENTS

Reviewer #1 (Remarks to the Author):

The paper by Maurano et al evaluates the functional consequences of SNVs at non-coding regions on altered DNA accessibility and predicted TF occupancy using a clever system of F1 hybrid mice. This system leverages the high genetic variability between mouse strains to identify a large number of heterozygous sites residing in differentially accessible DHSs, more that can be identified from human samples. The authors then use the data collected from four different tissues to quantify the effect size and cell-type specificity of thousands of SNVs throughout the mouse genome. In addition, they assign sensitivity profiles to TFs which is more efficient that performing allelic imbalance analysis from CHIP-seq data. The topic is of great significance to the functional genomics and population genetics fields. The paper is very well written and the data and analyses are of high quality. Therefore I recommend the paper for publication in Nature Communications.

Minor comments:

- 1) In the abstract the authors mention “all important TF families”. What do they mean by “important”? With many family members?
- 2) A paragraph at the end of the Introduction summarizing what will be presented in the paper would be beneficial.
- 3) References to figures 1g, 1h, 2e seem wrong (should be 1h, 1i, and 2b, respectively).
- 4) In figure 3a, the authors should include the genomic coordinates of the reference.
- 5) Supplementary Fig. 6 is not mentioned in the manuscript.

Reviewer #2 (Remarks to the Author):

Understanding how genetic variants, principally SNPs, influence the behavior of regulatory elements harbored within non-coding regions of the genome is an important pursuit, especially when it comes to the eventual appraisal of GWAS loci for human common complex traits. This current study, led by a highly established name in the field, sought to assess allelic differences in the context of DNA accessibility, via DNase I-hypersensitive mapping, and transcription factor binding leveraging five 'increasingly diverged strains of F1 hybrid mice'. This is an extremely attractive model in which to investigate such features, given that any variant on a particular chromosome is in perfect LD in the F1 offspring studied. This study represents crucial insight in to questions that have been posed in the human context, but have proved challenging to address systemically in that setting. Given the resulting data in made available, this represents an excellent resource for the community to refer. My concerns

are relatively minor:

1. In the final section of the Results, entitled 'Allelic effects on transcript levels', the authors are effectively conducting an eQTL study of sorts. This reviewer is not entirely convinced this aspect of the study is sufficiently statistically powered. The authors are therefore obligated to provide power calculations for this specific experiment.
2. Given the claims within the manuscript regarding the allelic relationship with TF occupancy, it is noted that no experimental validation is carried out. The authors need to delineate why they did not attempt any global ChIP-seq experiments for selected TFs in order to confirm their *in silico* observations.

Reviewer #3 (Remarks to the Author):

The manuscript submitted by Maurano et al. has provided an extensive description of the effect of non-coding variation in the mouse genome on chromatin accessibility measured using DNase hypersensitivity. The authors collect an impressive set of samples, in replicate, from multiple tissues harvested from F1 hybrid mice from five different crosses representing a range of mouse diversity among *Mus* species and subspecies. They document extensive allele-specific variation in accessibility dependent on tissue. The authors highlight that many of these variable accessible regions contain single nucleotide variants that occur within transcription factor motifs. While, the authors make the interesting observation highlighting the tissue-specific context of TF footprinting can be masked by aggregating across multiple cell types, tissue-specific regulatory activity has been well documented. In aggregate, these variable DHS hotspots show a very modest impact on nearby gene expression ($R < 0.2$), that is marginally detectable only because of their decreasing correlation based on distance to TSS.

The authors are to be applauded for their attention to detail in sample preparation, keen use of mouse genetics, and clearly high-quality data processing and analysis. The reviewer finds no fault to the technical aspects of the paper and appreciates the application of large-scale analysis to the mouse model organism. Some of the main conclusions of the paper, (i.e. extensive allele specificity in molecular activity due to cis-acting variants) has already been well documented in the literature. Including literature (appropriately) cited in the introduction for TF binding, chromatin, gene expression, and protein abundance. This leaves the reviewer curious to what, beyond a functional genomic resource, the current manuscript offers the larger field of understanding regulatory variation.

Generally, the authors would have benefited from editorial review of the manuscript, specifically in context of figure call outs and figure legends. This isn't simply a matter of convenience for reviewers as it gets increasingly frustrating when trying to interpret text that isn't necessarily easy to parse.

For example, page 4 first paragraph states, "We selected the highest-quality samples for deep paired-end...length distribution (Fig.1b). It really seems like that is not the data presented in figure 1b, but rather supplemental figure 1c?"

Nearly all the references to Figure 1 panels seem to be incorrect. I think Fig 1c (hierarchical clustering) actually is supposed to refer to Fig 1d, etc on down through the entire figure. And some parts of the figure (actual Fig 1b showing DHS profiles) don't seem to be referred to in the main text.

Another example...page 5 end of 2nd paragraph, "...the strongest single predictor was simply that the DHS be cell-type specific (Fig. 2e). There is no panel e in figure 2.

Last sentence page 5, "We found that aggregate imbalance was concentrated over the core positions of the motif...(Fig. 4c) – is that supposed to be 4d? To my eyes, it seems that 4c indicates enrichment within TF sequences, but not 'core positions'.

Page 6 first whole paragraph, "DNase I footprints (Fig. 5a)." Is not a sentence. What about them? I think this does refer to the correct panel, but there is no conclusion.

Discussion:

The primary claim made in the first sentence of the discussion is a bit of a circular argument. The authors state that, "Our work shows that most differences in DNA accessibility among mouse genomes can be attributed to direct perturbation of TF recognition sites." However, this claim is entirely dependent on the experimental setup and filtering process applied to the data. First, the authors only measure allele-specificity in F1 hybrids, therefore the only way they can potentially measure allelic differences is by the presence of SNVs used to genotype individual reads within DHSs. For example, if the authors were to measure differences in inbred parental lines first (as others have done), how many variable DHSs would they find in locations that lack variants? This is of particular importance in strains that are relatively closer in evolutionary history. Meaning B6x129 will have overall fewer SNVs between parental haplotypes, making it more difficult to accurately identify hotspots with variants in them, perforce not allowing the detection of significant differences between parental chromatin accessibility in F1s. I think what is driving this the underlying assumption from the outset is that all variation in DHS should be driven by cis-acting variants. While it is still true that, genome-wide, most variation in molecular features is still likely due in large part to cis-acting variants, that observation is also not necessarily new (Wittkopp, Haerum, and Clark, *Nature Genetics* 2007). There exists literature clearly indicating large-scale trans effects in chromatin (Rintisch et al. *Genome Research* 2014, Skelly et al. *Cell Stem Cell* 2020), gene expression (Tian Keller et al. *Genetics* 2015, Chesler et al. *Nature Genetics* 2005), or protein abundance (Chick et al. *Nature* 2016) in rodents. Genetically, cis- or trans- regulation is usually determined by comparing the bias in parents to allelic bias in F1 progeny to see if they correlate. Of course, large trans effects are less likely to be the case in human populations, which share much greater evolutionary similarity, but my understanding is that GTEx is indeed starting to identify abundant trans-regulation, at least in gene expression (GTEx Consortium, *Nature* 2017), which likely is driven by regulatory variation. The authors could fix this by moderating their claim to something along the lines of, "Our work shows that differences in DNA accessibility detected in hotspots with allelic imbalance identified through F1 hybrids can be attributed...". But it still doesn't circumvent the problem that they were only detected as being imbalance in F1s because they had variants in them.

What about sites that are highly variable, for which there are not SNVs? The approach taken here can not determine the extent to that type of regulator variation. While that might be outside the scope of the current manuscript, they claim that ‘most difference...among diverged mouse genomes’ seems too broad.

The authors state, “...we identify that a large proportion of imbalance can be linked to TF activity.” How much? (would be good to be clear in the discussion) Also, TF activity? or variation in TF binding? Are those the same?

Sometimes the authors switch between ‘DNase I footprint’ and ‘TF motif’. For example, in the discussion they state, “We expect that the enrichment of imbalanced SNVs in TF motifs observed in Fig. 4b...”. But 4b shows that there is an increase in imbalanced SNVs from ~2.4% to ~5.1% in DNase I footprints. Are those two terms so easily interchangeable?

The authors call attention to the fact that ‘nearly half’ of gained DHS represent co-option of existing DHS in new cell types. First, the ‘wholesale creation’ of DHS sites is biologically interesting, a shame to not discuss that further. Second, as this analysis was done in comparison to the reference genome (which is B6), it isn’t entirely clear the evolutionary role in this gain and loss. Is it ‘gained’ simply because B6 acquired a variant disrupting TF binding compared to the other strains? Why does it make sense that introduction of a new variant would co-opt a regulatory element already present in another tissue? Presumably those two regulatory elements are not being activated by the same cohort of TFs or the variant would impact binding equally between tissue. Or is that the interesting part of the observation? In which case it might be good to expand that discussion when talking about cooperative binding of TFs.

Results:

Some of the description of results are difficult to parse. This reviewer had difficulty understanding the claims and main points, in particular, surrounding figures 2 and 3. I understand that the analysis of gained and loss are in comparison to the reference genome. However, I think the authors miss an interesting opportunity to look at the evolutionary gain and loss of variable DHS sites. They have collected samples from a diverse set of *M.m.* subspecies, and they also collected data for *M. spret.* Using *M. spret.* as an outgroup, the authors could determine if the variants were shared with *spret.* (indicating evolutionary conservation) or a derived allele after divergence among the *M. musculus* lineages. While I understand this wasn’t the main goal of the manuscript, the use of gain and loss of sites could be confusing to readers who might think in evolutionary terms. As stated above, this would perhaps be of particular interest in comparing sites that were already DHS in other tissue.

Along those points, the authors conclude that, “point changes affecting accessibility act more frequently by broadening DNA accessibility...” This might be clarified by including “broadening DNA accessibility to other tissue...”.

The next sentence states, “from other lineages...”; because the authors data are representing both

mouse strain lineages, and difference cell types, perhaps changing this to “other cell lineages” will make it clearer.

The authors then go on to state, “...while co-option showed preference for related cell types, the strongest single predictor was simply that the DHS be cell-type specific”. Would be nice for the authors to provide an example of, “co-option showed preference for related cell types.” Can the authors point to the reader specifically where to go to get that observation?

“We identified clear examples of strong imbalance across multiple strains that was specific to a particular cell type (Fig. 3a)” This isn’t clear from the example. Does this refer to the left side of panel a? That shows a large imbalance in liver, a smaller one in lung, and no DHS activity in kidney or B-cells. Seemingly the lung imbalance doesn’t reach the 70% of reads mapping to one allele used as a filter for imbalance? Still looks like an imbalance in lung though. In my mind, the ideal candidate for this example would have high DHS signal in all four tissue, but only variable in one.

Next sentence. What do the authors mean by ‘morphology’ of a DHS? Read count? Again, not sure what part of panel a the authors refer to here, is it the right part of panel a? If so, I don’t see “...coordinate changes in...multiple nearby DHSs”. Perhaps a coordinate change in just the single nearest DHS though.

Next sentence. “Overall, however, we identified a higher degree of sharing of imbalance between samples of the same cell type than from the same strain or unrelated samples.” I can see the bar chart is highest for ‘same cell type’, but I’m having difficulty extracting biological meaning from that sentence. First, by ‘same strain’ do the authors mean same F1 hybrid progeny? Perhaps the authors mean that, as an example, same cell type would be in B6xCAST and B6xSPRET that liver DHSs show imbalance in the same direction, but that B6xCAST liver and B6xCAST kidney (in this case strain = (BxC)F1) show reduced imbalance in the same direction? That doesn’t seem all that surprising that tissue would behave similarly, particularly when you have every cross with the same maternal mouse strain (i.e. B6). Underlying that would probably just mean that a particular B6-specific variant is driving the shared imbalance in that tissue across multiple crosses. Going back to the evolutionary idea, likely that means that the reference genome (derived from B6) is likely harboring an alternative allele that diverged from the other wild-derived parental strains (which the authors state drive most of the imbalances).

Figures:

Figure 1: *Mus musculus* in reference to C57BL/6J, 129, and C3H should all be designated *Mus musculus domesticus*. Especially since you (correctly) indicate the subspecific nomenclature of *Mm musculus* and *Mm castaneus*.

Fig 2. What are the asterisks used to indicate? It is clear that they are the four tissues used in the study, but what do they represent here?

Fig 3. Panel a shows two different examples of DHS hotspots. Might make authors conclusions clearer to split these into two panels and try and highlight what the two different examples illustrate. As it is now,

it isn't clear. Also Panel a on the right, it is labeled ref(blue)/nonref(red) in the legend and with the read counts, but then the actual track profile shows that red is higher than blue (even though that doesn't agree with the raw read counts provided). So, are the colors incorrect or the legend? Additionally, the text underneath the 'blown up' track profiles is so tiny I can not read them when printed at all.

Fig 4. It isn't clear to this reviewer what the individual dots represent in panel c. Are these individual instances of motifs within a family? Per F1 hybrid? Per tissue?

Fig 5. It seems that the enrichments that are highlighted by outlines in panel b heat map are supposed to represent the examples in c-f, but it looks like CTF/NF-1 is highlighted in lung, but perhaps that is supposed to be AP-1, which is the row above? Panel d is AP-1 and seems enriched for imbalanced SNVs in lung. However, I don't actually see a signal for enrichment for lung in panel b (which is also not outlined). Also the legend says c-e but I think you intend it to be c-f.

Response to Reviewer Comments

We thank the Reviewers for their helpful comments and careful editorial attention. This has resulted in a number of key corrections and clarifications, changes to the text and figures, and several new analyses that confirm and strengthen the results of our manuscript. We have addressed specific comments in bold below.

REVIEWER COMMENTS

Reviewer #1 (Remarks to the Author):

The paper by Maurano et al evaluates the functional consequences of SNVs at non-coding regions on altered DNA accessibility and predicted TF occupancy using a cleaver system of F1 hybrid mice. This system leverages the high genetic variability between mouse strains to identify a large number of heterozygous sites residing in differentially accessible DHSs, more that can be identified from human samples. The authors then use the data collected from four different tissues to quantify the effect size and cell-type specificity of thousands of SNVs throughout the mouse genome. In addition, they assign sensitivity profiles to TFs which is more efficient that performing allelic imbalance analysis from ChIP-seq data. The topic is of great significance to the functional genomics and population genetics fields. The paper is very well written and the data and analyses are of high quality. Therefore I recommend the paper for publication in Nature Communications.

Minor comments:

1) In the abstract the authors mention “all important TF families”. What do they mean by “important”? With many family members?

Response: This was referring to the availability of motif models for historically well-studied TFs. We agree that “all important” overstates the case and have removed this clause from the abstract.

2) A paragraph at the end of the Introduction summarizing what will be presented in the paper would be beneficial.

Response: Thanks for the suggestion. We have added such a paragraph to the Introduction.

3) References to figures 1g, 1h, 2e seem wrong (should be 1h, 1i, and 2b, respectively).

Response: This has been corrected.

4) In figure 3a, the authors should include the genomic coordinates of the reference.

Response: We have added coordinates to Fig. 3a and 3b.

5) Supplementary Fig. 6 is not mentioned in the manuscript.

Response: A citation has been added (pg. 6). Note the figure has been renumbered Supplementary Fig. 7.

Reviewer #2 (Remarks to the Author):

Understanding how genetic variants, principally SNPs, influence the behavior of regulatory elements harbored within non-coding regions of the genome is an important pursuit, especially when it comes to the eventual appraisal of GWAS loci for human common complex traits. This current study, led by a highly established name in the field, sought to assess allelic differences in the context of DNA accessibility, via DNase I-hypersensitive mapping, and transcription factor binding leveraging five 'increasingly diverged strains of F1 hybrid mice'. This is an extremely attractive model in which to investigate such features, given that any variant on a particular chromosome is in perfect LD in the F1 offspring studied. This study represents crucial insight in to questions that have been posed in the human context, but have proved challenging to address systemically in that setting. Given the resulting data in made available, this represents an excellent resource for the community to refer. My concerns are relatively minor:

1. In the final section of the Results, entitled 'Allelic effects on transcript levels', the authors are effectively conducting an eQTL study of sorts. This reviewer is not entirely convinced this aspect of the study is sufficiently statistically powered. The authors are therefore obligated to provide power calculations for this specific experiment.

Response: In this section we claim that imbalanced DHSs show a correlation with allelic transcript levels that depends on distance to TSS. To assess the statistical significance of this effect, we permuted DNase allelic ratios among DHS-TSS pairs. The resulting 95% confidence band in Fig. 7 shows that the correlation between allelic accessibility and transcript levels is significant until it converges with background at long range (around ± 25 -100 kb). We note that this correlation reflects an aggregate analysis across all DHS/TSS, and we do not claim to have power to resolve individual DHS-TSS associations (i.e. individual eQTLs).

2. Given the claims within the manuscript regarding the allelic relationship with TF occupancy, it is noted that no experimental validation is carried out. The authors need to delineate why they did not attempt any global ChIP-seq experiments for selected TFs in order to confirm their in silico observations.

Response: Generation of global TF ChIP-seq data would have considerably increased the scale of the project while covering only a limited selection of TFs. Indeed this is a key advantage of our DNase-seq based strategy, which reflects our past experience progressing from ChIP-seq to DNase-seq (Maurano et al PLoS Genetics 2012 and Nat. Genetics 2015).

To provide a completely independent validation dataset and minimize additional animal work, we turned to a mESC line generated from B6xCAST mice. We have generated new DNase-seq (n=2) and ChIP-seq (n=1) datasets (see the new Supplementary Table 5) and processed them using our pipeline for allelic analysis. The new Supplementary Fig. 6 confirms that CATO2 in silico predictions of functional noncoding variation demonstrate undiminished predictive power on the new independent validation datasets.

Reviewer #3 (Remarks to the Author):

The manuscript submitted by Maurano et al. has provided an extensive description of the effect of non-coding variation in the mouse genome on chromatin accessibility measured using DNase hypersensitivity. The authors collect an impressive set of samples, in replicate, from multiple tissues harvested from F1 hybrid mice from five different crosses representing a range of mouse diversity among *Mus* species and subspecies. They document extensive allele-specific variation in accessibility dependent on tissue. The authors highlight that many of these variable accessible regions contain single nucleotide variants that occur within transcription factor motifs. While, the authors make the interesting observation highlighting the tissue-specific context of TF footprinting can be masked by aggregating across multiple cell types, tissue-specific regulatory activity has been well documented. In aggregate, these variable DHS hotspots show a very modest impact on nearby gene expression ($R < 0.2$), that is marginally detectable only because of their decreasing correlation based on distance to TSS.

Response: We thank the Reviewer for the positive comments. Regarding the low impact on gene expression, it is important to note that this is a global comparison, considering all possible DHS-TSS pairs. We would expect the aggregate correlation could increase with development of more specific methods to assign target genes. We have updated the Discussion to cover this important point:

“The global correlation observed between allelic accessibility and allelic transcript levels was statistically significant but modest. Much as the majority of point variants are buffered in terms of their effect on local chromatin features⁵, enhancer networks controlling gene expression likely demonstrate a high degree of redundancy and selectivity^{25,32,33}. The correlation we observe could serve as a benchmark for development of genome-wide methods to predict likely target genes of distal regulatory elements, and complements systematic locus-scale investigation of regulatory architecture using genome engineering^{33,34}. Thus it is likely that further exploitation of mouse genetics will provide the substrate for more granular models of enhancer-promoter interaction.”

The authors are to be applauded for their attention to detail in sample preparation, keen use of mouse genetics, and clearly high-quality data processing and analysis. The reviewer finds no fault to the technical aspects of the paper and appreciates the application of large-scale analysis to the mouse model organism. Some of the main conclusions of the paper, (i.e. extensive allele specificity in molecular activity due to cis-acting variants) has already been well documented in the literature. Including literature (appropriately) cited in the introduction for TF binding, chromatin, gene expression, and protein abundance. This leaves the reviewer curious to what, beyond a functional genomic resource, the current manuscript offers the larger field of understanding regulatory variation.

Response: We thank the Reviewer for their positive assessment. While it is true that cell-type specific genetic effects on gene regulation are expected based on a textbook reading, few studies have had power to assess them genome-wide. Combined with our approach to employ mouse genetics to assess human cis-regulatory variation, we hope that our work leads the way for expanded use of model organisms to inform human genetics.

Generally, the authors would have benefited from editorial review of the manuscript, specifically in context of figure call outs and figure legends. This isn't simply a matter of convenience for reviewers as it gets increasingly frustrating when trying to interpret text that isn't necessarily easy to parse.

Response: We thank the Reviewers for their detailed attention to our manuscript and apologize to for the errors that complicated review. In addition to the close reading by all three Reviewers , we have thoroughly edited and proof-read the manuscript. We have made numerous corrections and clarifications throughout the figures and Discussion which we hope clarify the novelty and impact of our work.

For example, page 4 first paragraph states, "We selected the highest-quality samples for deep paired-end...length distribution (Fig.1b). It really seems like that is not the data presented in figure 1b, but rather supplemental figure 1c?

Response: Fig. 1c shows the fragment length distribution for the corresponding samples in Fig. 1b. We have added a citation of Supplementary Fig. 1c, which shows the fragment-length distributions for all samples.

Nearly all the references to Figure 1 panels seem to be incorrect. I think Fig 1c (hierarchical clustering) actually is supposed to refer to Fig 1d, etc on down through the entire figure. And some parts of the figure (actual Fig 1b showing DHS profiles) don't seem to be referred to in the main text.

Another example...page 5 end of 2nd paragraph, "...the strongest single predictor was simply that the DHS be cell-type specific (Fig. 2e). There is no panel e in figure 2.

Response: These citations have been corrected (see also Reviewer 1, point 3).

Last sentence page 5, "We found that aggregate imbalance was concentrated over the core positions of the motif...(Fig. 4c) – is that supposed to be 4d? To my eyes, it seems that 4c indicates enrichment within TF sequences, but not 'core positions'.

Response: The citation of Fig. 4c is correct: the enrichment on the x-axis of Fig. 4c refers to the rate of imbalance in recognition sequences versus flanking regions (shaded gray in Fig. 4d). We have updated the x-axis label to clarify.

Page 6 first whole paragraph, "DNase I footprints (Fig. 5a)." Is not a sentence. What about them? I think this does refer to the correct panel, but there is no conclusion.

Response: This sentence fragment has been corrected:

"We found higher rates of cell-type specific imbalance at sites of DNase I footprints in matching cell and tissue types, relative to unmatched cell and tissue types (Fig. 5a)."

Discussion:

The primary claim made in the first sentence of the discussion is a bit of a circular argument. The authors state that, "Our work shows that most differences in DNA accessibility among mouse genomes can be attributed to direct perturbation of TF recognition sites." However, this claim is entire dependent on the experimental setup and filtering process applied to the data. First, the authors only measure allele-specificity in F1 hybrids, therefore the only way they can potentially measure allelic differences is by the presence of SNVs used to genotype individual reads within DHSs. For example, if the authors were to measure differences in inbred parental lines first (as others have done), how many variable DHSs would they find in locations that lack variants?

This is of particular importance in strains that are relatively closer in evolutionary history. Meaning B6x129 will have overall fewer SNVs between parental haplotypes, making it more difficult to accurately identify hotspots with variants in them, perforce not allowing the detection of significant differences between parental chromatin accessibility in F1s. I think what is driving this the underlying assumption from the outset is that all variation in DHS should be driven by cis-acting variants. While it is still true that, genome-wide, most variation in molecular features is still likely due in large part to cis-acting variants, that observation is also not necessarily new (Wittkopp, Haerum, and Clark, Nature Genetics 2007). There exists literature clearly indicating large-scale trans effects in chromatin (Rintisch et al. Genome Research 2014, Skelly et al. Cell Stem Cell 2020), gene expression (Tian Keller et al. Genetics 2015, Chesler et al. Nature Genetics 2005), or protein abundance (Chick et al. Nature 2016) in rodents. Genetically, cis- or trans- regulation is usually determined by comparing the bias in parents to allelic bias in F1 progeny to see if they correlate. Of course, large trans effects are less likely to be the case in human populations, which share much greater evolutionary similarity, but my understanding is that GTEx is indeed starting to identify abundant trans-regulation, at least in gene expression (GTEx Consortium, Nature 2017), which likely is driven by regulatory variation. The authors could fix this by moderating their claim to something along the lines of, “Our work shows that differences in DNA accessibility detected in hotspots with allelic imbalance identified through F1 hybrids can be attributed...”. But it still doesn’t circumvent the problem that they were only detected as being imbalance in F1s because they had variants in them.

What about sites that are highly variable, for which there are not SNVs? The approach taken here can not determine the extent to that type of regulator variation. While that might be outside the scope of the current manuscript, they claim that ‘most difference...among diverged mouse genomes’ seems too broad.

Response: The Reviewer correctly notes that there are several factors which impact the ability to detect allelic imbalance. Fig. 1h. confirms that while the more diverged strains (B6xPWK, B6xCAS, and B6xSPRET) have at least one SNV in the majority of DHSs, B6x129 and B6xC3H are more limited by sequence diversity.

Our goal was to survey cis-regulatory variants, thus we were able to generate data for double the number of cell/tissue types or strains by focusing solely on F1 hybrids than if we had additionally profiled each parental line. An important conclusion of our work is that the mappability challenges posed by the more diverged genomes are surmountable, and thus power can be maximized by expanding profiling to new cell and tissue types using the most diverged subspecies.

The Reviewer points out that our wording could be interpreted in an overly expansive fashion. Therefore we have edited that line in the Discussion to limit our conclusions to cis-regulatory variants. We also have added a mention of trans regulation to the Discussion, citing some of the references suggested above:

“It would also be straightforward to incorporate trans-regulatory differences between strains or species into future models to enable analysis of trans-regulatory effects on gene expression^{29-31”}

The authors state, “...we identify that a large proportion of imbalance can be linked to TF activity.” How much? (would be good to be clear in the discussion)

Response: This refers to the proportion of imbalanced sites overlapping TF motifs at a reasonable cutoff. We have added the exact percentage to the sentence in the Discussion:

“By analyzing a broad set of TFs with known sequence specificities, we identify that fully 61% of imbalanced sites can be linked to TF activity (Fig. 4a).”

Also, TF activity? or variation in TF binding? Are those the same?

Response: We have removed “TF activity” and replaced it with “TF occupancy” or other wording as appropriate.

Sometimes the authors switch between ‘DNase I footprint’ and ‘TF motif’. For example, in the discussion they state, “We expect that the enrichment of imbalanced SNVs in TF motifs observed in Fig. 4b...”. But 4b shows that there is an increase in imbalanced SNVs from ~2.4% to ~5.1% in DNase I footprints. Are those two terms so easily interchangeable?

Response: The Reviewer correctly notes that, although DNase I footprints and TF motif matches overlap more frequently than expected by chance (Neph et al., Nature 2012), they are distinct classes of genomic elements. Fig. 4b includes both as independent factors and shows two effects: a general increase in imbalance relative to FIMO P-value thresholds (TF motifs), and a further increase when comparing within each FIMO P-value bin motifs that have DNase I footprints to those without. We have clarified this point in the figure legend.

The authors call attention to the fact that ‘nearly half’ of gained DHS represent co-option of existing DHS in new cell types. First, the ‘wholesale creation’ of DHS sites is biologically interesting, a shame to not discuss that further. Second, as this analysis was done in comparison to the reference genome (which is B6), it isn’t entirely clear the evolutionary role in this gain and loss. Is it ‘gained’ simply because B6 acquired a variant disrupting TF binding compared to the other strains? Why does it make sense that introduction of a new variant would co-opt a regulatory element already present in another tissue? Presumably those two regulatory elements are not being activated by the same cohort of TFs or the variant would impact binding equally between tissue. Or is that the interesting part of the observation? In which case it might be good to expand that discussion when talking about cooperative binding of TFs.

Response: We agree this is one of the key findings of our work that will inspire future studies. We have expanded upon this point in the Discussion:

“We speculate that functional regulatory variation might most easily arise from creation of TF recognition sequences that expand the selectivity of an existing DHS by acting cooperatively with existing TF recognition sequences.”

Results:

Some of the description of results are difficult to parse. This reviewer had difficulty understanding the claims and main points, in particular, surrounding figures 2 and 3. I understand that the analysis of gained and loss are in comparison to the reference genome. However, I think the authors miss an interesting opportunity to look at the evolutionary gain and loss of variable DHS sites. They have collected samples from a diverse set of M.m. subspecies, and they also collected data for M. spret. Using M spret as an outgroup, the authors could determine if the variants were shared with spret (indicating evolutionary conservation) or a derived allele after divergence among the M musculus lineages. While I understand this wasn’t the main goal of the manuscript, the use of gain and loss of sites could be confusing to readers who might think in evolutionary terms. As stated above, this would perhaps be of particular interest in comparing sites that were already DHS in other tissue.

Response: The Reviewer points out that “gain” and “loss” imply an unexplored evolutionary argument which is secondary to our focus. We have changed the manuscript to instead use “reference higher” and “non-reference higher”.

Along those points, the authors conclude that, “point changes affecting accessibility act more frequently by broadening DNA accessibility...” This might be clarified by including “broadening DNA accessibility to other tissue...”.

Response: This has been clarified (pg. 4):

“This suggests that point changes affecting accessibility at sites with preexisting activity act more frequently by broadening DNA accessibility to other cell and tissue types, rather than de novo evolution of novel regulatory DNA”.

The next sentence states, “from other lineages...”; because the authors data are representing both mouse strain lineages, and difference cell types, perhaps changing this to “other cell lineages” will make it clearer.

Response: Thank you for the helpful suggestion – this sentence has been clarified:

“This cell-type specific expansion of DHS activity drew broadly from other cell lineages...”

The authors then go on to state, “...while co-option showed preference for related cell types, the strongest single predictor was simply that the DHS be cell-type specific”. Would be nice for the authors to provide an example of, “co-option showed preference for related cell types.” Can the authors point to the reader specifically where to go to get that observation?

Response: We have clarified in the main text and legend that the samples along the y-axis of Fig. 2b are hierarchically clustered based on genome-wide DHSs. Thus related cell and tissue types show up near the cognate tissues, which are now more clearly highlighted with blue triangles.

“We identified clear examples of strong imbalance across multiple strains that was specific to a particular cell type (Fig. 3a)” This isn’t clear from the example. Does this refer to the left side of panel a? That shows a large imbalance in liver, a smaller one in lung, and no DHS activity in kidney or B-cells. Seemingly the lung imbalance doesn’t reach the 70% of reads mapping to one allele used as a filter for imbalance? Still looks like an imbalance in lung though. In my mind, the ideal candidate for this example would have high DHS signal in all four tissue, but only variable in one.

Response: That sentence actually referred to both example sites. To clarify, we have split Fig. 3a into separate panels a and b (see later point also referring to Fig. 3a). The Reviewer is correct that the example on the left (now Fig. 3a) shows a DHS active in liver and lung but with imbalance only in liver. Lung shows a nearly 50:50 distribution of reads reflected in the even split between red and blue; we have clarified by adding an asterisk to denote significantly imbalanced cell or tissue types.

The example on the right (now Fig. 3b) shows a strong DHS in all 4 tissues but only significantly imbalanced in B cells. We have added this interpretation to the figure legend.

Next sentence. What do the authors mean by ‘morphology’ of a DHS? Read count? Again, not sure what part of panel a the authors refer to here, is it the right part of panel a? If so, I don’t see “...coordinate changes in...multiple nearby DHSs”. Perhaps a coordinate change in just the single nearest DHS though.

Response: “Morphology” meant to refer to the two adjacent peaks in a composite DHS in Fig. 3a. We have simplified the text:

“In both examples, cell-type specific imbalance in one DHS was associated with a coordinate change in accessibility at a nearby DHS (Fig. 3a-b), though we note that it is not possible to infer the direction of causality.”

Next sentence. “Overall, however, we identified a higher degree of sharing of imbalance between samples of the same cell type than from the same strain or unrelated samples.” I can see the bar chart is highest for ‘same cell type’, but I’m having difficulty extracting biological meaning from that sentence. First, by ‘same strain’ do the authors mean same F1 hybrid progeny? Perhaps the authors mean that, as an example, same cell type would be in B6xCAST and B6xSPRET that liver DHSs show imbalance in the same direction, but that B6xCAST liver and B6xCAST kidney (in this case strain = (BxC)F1) show reduced imbalance in the same direction?

Response: The Reviewer’s interpretation of the categories is correct. “Same strain” serves as a more conservative negative control than “Unrelated” samples which potentially accounts for trans effects or the influence of linked variants. We have clarified by coloring these two measurements gray in Fig. 3c. The correlation among “Same cell/tissue” type is significantly above both of the background comparisons, confirming that cell/tissue-type genetic effects dominate and paralleling a convergent result based solely on DHSs in Fig. 1d.

That doesn’t seem all that surprising that tissue would behave similarly, particularly when you have every cross with the same maternal mouse strain (i.e. B6). Underlying that would probably just mean that a particular B6-specific variant is driving the shared imbalance in that tissue across multiple crosses. Going back to the evolutionary idea, likely that means that the reference genome (derived from B6) is likely harboring an alternative allele that diverged from the other wild-derived parental strains (which the authors state drive most of the imbalances).

Response: Supplementary Fig. 3 shows that the overall set of imbalanced SNVs is equally split between reference (i.e. C57BL/6J) and non-reference, thus Fig. 3c reflects the effects of both reference and nonreference alleles.

Figures:

Figure 1: Mus musculus in reference to C57BL/6J, 129, and C3H should all be designated Mus musculus domesticus. Especially since you (correctly) indicate the subspecific nomenclature of Mm musculus and Mm castaneus.

Response: Thanks for catching this. Fig. 1a has been corrected to include “domesticus” for C57BL/6J, 129S1/SvImJ, and C3H/HeJ.

Fig 2. What are the asterisks used to indicate? It is clear that they are the four tissues used in the study, but what do they represent here?

Response: The asterisks in Fig. 2b highlighted the ENCODE reference DNase-seq datasets best matching the tissues from hybrid mice in this study. We have changed them to more prominent triangles and shifted them underneath the specific hybrid cell/tissue type columns to clarify.

Fig 3. Panel a shows two different examples of DHS hotspots. Might make authors conclusions clearer to split these into two panels and try and highlight what the two different examples illustrate. As it is now, it isn’t clear. Also Panel a on the right, it is labeled ref(blue)/nonref(red) in the legend and with the read counts, but then the actual track profile shows that red is higher than blue (even though that doesn’t agree with the raw read counts provided). So, are the colors incorrect or the legend? Additionally, the text underneath the ‘blown up’ track profiles is so tiny I can not read them when printed at all.

Response: We have split Fig. 3a into separate panels a and b as suggested, enlarged the figure text, and clarified both the figure and its legend. Indeed, in Fig. 3b the colors (representing the proportion of the overall read density attributable to each allele based on informative SNPs) had been inadvertently reversed. We apologize for the confusion.

Fig 4. It isn't clear to this reviewer what the individual dots represent in panel c. Are these individual instances of motifs within a family? Per F1 hybrid? Per tissue?

Response: We have updated Fig. 4c and its legend to clarify that the individual dots correspond to a given TF motif, which are grouped into TF families on the Y-axis. Data in Fig. 4 are aggregated over all strains and cell/tissue types.

Fig 5. It seems that the enrichments that are highlighted by outlines in panel b heat map are supposed to represent the examples in c-f, but it looks like CTF/NF-1 is highlighted in lung, but perhaps that is supposed to be AP-1, which is the row above? Panel d is AP-1 and seems enriched for imbalanced SNVs in lung. However, I don't actually see a signal for enrichment for lung in panel b (which is also not outlined). Also the legend says c-e but I think you intend it to be c-f.

Response: We have updated the outlines in Fig. 5b to correspond to panels c-f. AP-1 is enriched in lung (as shown in panel c), albeit at a lower level than the other examples. We have adjusted the color scale to make this clearer. We have corrected the panel labels in the legend.

REVIEWERS' COMMENTS

Reviewer #2 (Remarks to the Author):

The authors have satisfied the concerns of this reviewer. Great study!

Reviewer #3 (Remarks to the Author):

I appreciate the consideration the authors gave my review. I find the changes clarify any early confusion and generally the manuscript is improved. I think there are further aspects of this work that remain interesting for future questions and I appreciate these are beyond the scope of the current study. While I approve of the authors claims in the discussion, I still find that the ~40% of cis-linked differences in accessibility that cannot currently be linked to perturbation of TF sites to be interesting. Perhaps that is artificially high due to statistical cutoffs and incomplete knowledge of TFs. Regardless, the work stands on its own as is.